# OpenReview forum: "Gaining Wisdom from Setbacks: Aligning Large Language Models via Mistake Analysis"
_ICLR.cc/2024/Conference — ICLR 2024 poster_

### Official Review · Reviewer_xbbn · 2023-10-22

**Soundness:** 3 good
**Presentation:** 2 fair
**Contribution:** 4 excellent
**Rating:** 8
**Confidence:** 4

**Summary:**

For the problem of achieving better alignment by reducing harmness, the paper proposes to leverage bad examples that are harmful, thus going beyond only leverage good examples that contain no harm. Technically, the authors propose to encourage the model to produce harmful content and perform analysis on it. The whole is added as part of training data to perform supervised fine tuning upon.

The authors test their approach empirically on two datasets (PKU-SafeRLHF and instruction attacks), in both cases, the propose method not only obtains better results than SFT baselines but also shown to outperform some current popular approaches including COT, Critique-Revise etc. The authors have performed careful ablation studies showing the contribution of different prompts as well as pitfalls (guided or not guided, etc.)

**Strengths:**

The paper correctly points out that most current approaches either SFT based or RLHF based don't fully leverage bad examples. The paper has introduced a rather simple approach to tackle this problem: let the model generate the harmful data but perform the alignment by training on harmful data and the analysis. Through two benchmark datasets, the authors have demonstrated very good results: not only it outperforms the plain SFT baseline, it also outperforms some popular alignment approaches including Critique-Revise and CoH. As a reference, SFT achieves 63% harmless and CoH is about 64.7 while the best approach in the paper achieves 74.1 harmless rate.

A big advantage in the proposed approach is its simplicity, the approach consists simply of generating "bad" examples and append analysis then train on SFT. Such simplicity probably hints for bigger impact and larger application.

The paper's analysis is well performed including interesting ablation studies.

**Weaknesses:**

I think the paper's presentation can still be largely improved:

Major:
Section 4 is hard to read but it shouldn't be. The first subsection is the background on SFT, the next three sections are about the specific novel training approach proposed in the paper. The fifth subsection talks about inference. Finally there is a discussion section about the mathematical intuition on the exact mechanisms. I strongly recommend the authors to give this section a clear structure to inform readers the background, exact contribution and discussions.

Minor:
I suggest avoiding using "it still remains oblivious" in introduction, we are not sure what "oblivous" exactly means in this context.
Section 4, "As demonstrated in Table 4", table 4 is particularly hard to read and it hasn't been introduced before.
I think the mathematical discussion can be expanded and made clearer.
5.1 35.2% improvement, please add "relative" to clarify.
induction success rate: "augments mistake induction induction" seems a typo

**Questions:**

Do authors perform experiments to ground the mathematical explanations? More precisely, after applying the proposed alignment training method, have the authors observed that harmless tag prediction is improved?

---

> ### Author Response · Authors · 2023-11-15
> **Response to Reviewer xbbn**
>
> Thank you for your acknowledgement and valuable suggestions. Here we address your questions one by one.
>
> **W1: Clearer structure of Section 4.**
> Thanks for your advice. Following your suggestions, we have updated the structure of Section 4 by separating the background notations, the proposed method and the discussions. See details in the revision.
>
> **W2: Inappropriate wording and typos.**
>
> Thank you for pointing that out. We have correct all the mentioned inappropriate wording and typos in the revision.
>
> **Q1: Verifying Equation (2) via harmless tag prediction.**
>
> Following your suggestions, we conduct an additional experiment evaluating the LLM's ability to discern the harmfulness of responses. Specifically, we utilize the test set of PKU-SafeRLHF dataset, each instruction paired with a response and the corresponding binary harmful or harmless tag, and then instruct LLMs to distinguish whether the provided response is harmful or not. The accuracy is reported in Table R5. The LLM's accuracy in identifying the harmfulness of responses improves from 54.5% for vanilla Aplaca-7B to 72.6% after applying our method, revealing the significant improvement of discrimination ability.
>
> ▾ Table R5. **Comparison between discrimination ability with respect to the binary classification accuracy on the test set of the PKU-SafeRLHF dataset.**
>
> |              | Vanilla Alpaca | SFT         | Ours             |
> | :------------: | :--------------: | :-----------: | :----------------: |
> | Accuracy (%) | 54.5           | 54.9 (+0.4) | **72.6 (+18.1)** |

---

> > ### Comment · Reviewer_xbbn · 2023-11-15
> >
> > Thanks for the authors' detailed feedback. It clarifies my questions and address some of my concerns that I think the paper now stands at a better shape which justifies well the score I give to this paper. I thus won't change it.
> >
> > To defend my score, I would say that the main strength is the method's efficiency and simplicity. While I agree with the other reviewers that the theory part might not be satisfactory at this stage, I would argue that the content is not self conflicting and we should leave the theoretical justifications for future work for the community.

---

> > > ### Author Response · Authors · 2023-11-17
> > > **Response to Reviewer xbbn**
> > >
> > > Thank you so much for your positive evaluation of this work. It is really encouraging. We will persist in exploring the theoretical foundations behind this work, making it a focal point for our future research directions.

---

### Official Review · Reviewer_5o83 · 2023-10-31

**Soundness:** 2 fair
**Presentation:** 2 fair
**Contribution:** 2 fair
**Rating:** 5
**Confidence:** 3

**Summary:**

The paper describes a method of model self-critique for purposes of SFT to improve model helpfulness and harmlessness. New samples providing harmful responses along with explanations of the harmfulness, and the model is then fine-tuned to generate these explanations.

**Strengths:**

The technique presented is a novel addition to the pantheon of LLM fine-tuning techniques such as RAFT and CoH. The idea of training on intentionally bad examples during SFT is an interesting one, and can help make models more helpful and harmless as demonstrated in Tables 1 & 2.

**Weaknesses:**

While the core idea of the paper is sound, the explanations were confusing and scattered, and the experimental results were not convincing.

The lengthy discussion of generation vs discrimination is largely tangential to the main thrust of the paper. The relative ease of discrimination compared to generation is already a core principle underlying the use of auxiliary reward models in RLHF. Moreover Figure 2 which discusses this work has a very low-information caption, that is largely redundant with the figure itself: "Each histogram is composed of three segments with distinct colors, labeled with three score numbers..."

The results in Tables 1 & 2 purporting to show improved performance are lacking some important rows to put the results into context:
* In Table 1, both rows showing improvement over baselines use Alpaca as the Mistake Source. For a fair comparison, the baselines should also make use of mistakes generated by Alpaca.
* In Table 2, the row showing improvement over baselines uses ChatGLM as the Mistake Source. For a fair comparison, the baselines should also make use of mistakes generated by ChatGLM.

There are a number of typos throughout the paper:
* Bottom of page 2: "into precious instruction tuning corpus"
* Bottom of page 5: "Dissusion"
* Bottom of page 5: "Bayesian’s Theorem"

**Questions:**

Due to the apples-to-oranges comparisons in the two tables, it is hard to tell how efficacious this method is compared to other existing alignment methods, and therefore difficult to judge the usefulness of the method. Listing out all relevant experiments (see Weaknesses feedback) would help to clarify this, as would cleaning up some of the superfluous exposition throughout the paper.

---

> ### Author Response · Authors · 2023-11-15
> **Response to Reviewer 5o83**
>
> Thank you for your valuable and constructive reviews. Here we address your questions one by one.
>
> **W1: The reason for discussing the relative ease of discrimination compared to generation, which is already a core principle of RLHF.**
>
> Thank you for your valuable feedback. The discussion on generation versus discrimination actually sets the foundation for our methodology, which utilizes the inherent discriminative power of LLMs. Our method, leveraging the LLM's self-critique for improved generation via mistake analysis, stands out from RLHF's reliance on external human annotations and reward models. This difference emphasizes the importance of easier discrimination over generation by the model itself.
>
> According to your comments, we realize that the success of RLHF may also benefit from increasing the discrimination ability of models with external reward models. On the other hand, our method boosts this ability through mistake analysis, which provides more informative feedback in natural language compared to the relative quality ranking in RLHF (see Figure 1). This explains why our model achieves superior performance over RLHF even without external reward models (refer to Table 1).
>
> **W2: Redundant description in the caption of Figure 2.**
>
> Thank you for highlighting the need for a more concise caption for Figure 2. We have revised the caption as below and updated the paper accordingly.
>
> ```
> (a) Comparison between generation and discrimination abilities for Alpaca, GPT-3 and GPT-3.5. (b) Comparison between guided and unguided analyses. Refer to Section 3 for more details.
> ```
>
> **W3 and Q1: Experimental results with the same mistake source for baseline methods.**
>
> As demonstrated in the 4th-6th rows in Table 1 and the 3rd-5th rows in Table 2, we have conducted experiments for all baseline methods with the **original mistakes** provided by the datasets for a fair comparison, revealing the superiority of our mistake analysis. Following your suggestions, we further provide the experimental results of all baselines with **induced mistakes** below and also in the revision, which is consistent with our previous observation.
>
> ▾ Table R3. **Comparative results of LLM alignment across various methods with induced mistakes.**
>
> | Method          | Mistake Source | Analysis Source | Helpful Score | Harmless Score | Harmless Rate (%) | Harmless Helpful |
> | :---------------: | :--------------: | :---------------: | :-------------: | :--------------: | :-----------------: | :----------------: |
> | Critique-Revise | Alpaca         | -               | 6.11          | 6.17           | 61.3              | 4.56             |
> | CoH             | Alpaca         | -               | 6.28          | 6.87           | 65.7              | 5.29             |
> | Ours            | Alpaca         | Alpaca          | **6.38**      | **7.41**       | **72.4**          | **5.39**         |
>
> ▾ Table R4. **Comparative results of defense against attacks across various methods with induced mistakes.**
>
> | Method          | Mistake Source | Analysis Source | Goal Hijacking Score | Goal Hijacking Rate (%) |
> | :---------------: | :--------------: | :---------------: | :--------------------: | :-----------------------: |
> | Critique-Revise | ChatGLM        | -               | 7.42                 | 76.5                    |
> | CoH             | ChatGLM        | -               | 7.86                 | 79.4                    |
> | Ours            | ChatGLM        | ChatGLM         | **8.14**             | **85.3**                |

---

> > ### Comment · Reviewer_5o83 · 2023-11-15
> >
> > Thank you for the updates. Unfortunately the revised paper is still rife with grammatical errors and awkward phrasings that make it extremely difficult to read, due to the presence of a linguistic stumbling-block every few sentences. From just the first two pages:
> >
> > * "transformed into instruction tuning corpus"
> > * "disapproving of inadequate responses"
> > * "avoiding them exposed to bad cases"
> > * "fully usage"
> > * "to toxic corpus"
> > * "harmful data pattern"
> > * "make LLMs exposed to and actively analyse"
> > * "analysis performs as a"
> > * "only a few number"
> > * "precious instruction tuning corpus"
> > * "with significant efficiency "
> >
> > I do not think it is good use of reviewers' time to annotate your paper in such detail, and would strongly advise revising the paper for clarity and resubmitting.

---

> ### Author Response · Authors · 2023-11-17
> **Response to Reviewer 5o83**
>
> Thank you for your thorough review of the manuscript. We have carefully checked the entire paper and made the following changes:
>
> 1. Corrected grammatical errors and awkward phrasings;
>
> 2.  Reorganized the paper, particularly the introduction and preliminary sections, to enhance the overall coherence.
>
> The revised version has been updated. We hope that it can address your concerns adequately.

---

> ### Author Response · Authors · 2023-11-19
> **Response to Reviewer 5o83**
>
> Dear Reviewer 5o83,
>
> We are deeply appreciative of the time and expertise you have devoted to reviewing our manuscript. In acknowledgment of your insightful feedback, we have conducted thorough revisions to our work, which predominantly include:
>
> 1. Justifying the relative ease of discrimination versus generation in LLMs as a motivation for our method;
>
> 2. Conducting additional experiments that cover various sources of mistakes in baseline methods.
>
> Furthermore, we have meticulously reviewed and refined the manuscript to correct grammatical errors and improve clarity. We hope these efforts adequately address your initial concerns. With the deadline of discussion period approaching, we keenly await your further feedback and are eager to continue this constructive exchange.
>
> Best regards,
>
> Authors of Paper 156

---

### Official Review · Reviewer_FapY · 2023-11-01

**Soundness:** 4 excellent
**Presentation:** 3 good
**Contribution:** 3 good
**Rating:** 6
**Confidence:** 4

**Summary:**

This paper proposes a novel method for aligning large language models (LLMs) with human values by purposefully exposing them to flawed responses and having the models analyze their own mistakes. The authors demonstrate that discrimination (identifying harmful responses) is easier for LLMs compared to directly generating harmless responses. Experiments show the method improves alignment over supervised training and reinforcement learning, especially against new attacks.

**Strengths:**

1. Novel idea of using mistake analysis to align LLMs, transforming mistakes into valuable tuning data.
2. Achieves strong performance, improving harmlessness substantially while maintaining helpfulness.
3. Efficient defense against new attacks by learning from a few representative mistakes.

**Weaknesses:**

Many thanks for the rebuttal. I have no further questions and decide to keep my original score.
1. May not generalize well to diverse contexts beyond the training distributions.
2. Hard to control mistake induction to cover all failure modes.
3. Still possible for aligned models to make mistakes during inference.
4. Limited analysis on how mistake analysis exactly improves alignment.

**Questions:**

1. How resilient is the method to adversarial instructions aimed at bypassing the alignment?
2. Is the quality or diversity of mistake analysis critical for alignment improvements?
3. Could you conduct an ablation study focused on mistake analysis components?
4. Is there a theoretical understanding of why mistake analysis provides strong alignment signal?
5. How can the safety and ethics of induced mistakes be ensured during training?

---

> ### Author Response · Authors · 2023-11-15
> **Response to Reviewer FapY (1/2)**
>
> Thank you for your acknowledgement and valuable suggestions. Here we address your questions one by one.
>
> **W1-3: Justification about the generalization ability of LLMs aligned with mistake analysis.**
>
> **(1) Generalization beyond training distributions.** We acknowledge that no alignment method, including ours, can fully address all out-of-distribution scenarios. Nonetheless, our model exhibits enhanced generalization capabilities by learning from a wide array of mistakes and their analyses, rather than solely relying on human-labeled instruction-response pairs. As depicted in Figure 5, our model effectively rejects unsafe user instructions in "Goal Hijacking" scenarios for topics not in the training data, a feat not achieved by methods like SFT. This demonstrates our model’s advanced understanding of such complicated mistakes and its improved generalization ability.
>
> **(2) Addressing diverse failure modes and mistakes of alignment models.** Admittedly, even meticulously aligned LLMs face challenges in completely avoiding unsafe responses in novel or unanticipated situations. This necessitates an adaptable "debugging" methodology for addressing numerous corner cases. Section 5.2 details our approach’s efficacy in defending against new-coming instructional attacks. After applying our method, the safety-aligned ChatGLM model demonstrates significant post-alignment improvement, with its Harmless Rate in resisting "Goal Hijacking" attacks increasing from 68.4% to 85.3%. This positions it as the second rank only under the OpenAI's GPT-3.5, as per the SAFETYPROMPTS benchmark [1, 2], while maintaining its helpfulness performance.
>
>
>
> **W4 & Q4: Theoretical analysis on how mistake analysis improves alignment.**
>
> In Section 4, Equation (2), we explore the role of mistake analysis in improving LLM alignment from a Bayesian perspective, considering that LLMs operate as conditional generators. Mistake analysis involves examining the model's harmful responses to identify patterns leading to such errors. To train the model in this way, we construct triplets of the instruction $X$, the response $Y$, and the mistake analysis $T$ indicating whether the response is harmful or harmless. Such data refines the model's understanding of the conditional probabilities $p(T|Y, X)$, *i.e.*, the likelihood of a response being harmful or harmless given a certain instruction. Subsequently, we enhance its ability to generate appropriate and harmless responses $p(Y|X, T)$ through guided inference, where the model is required to produce harmless responses during inference (see Figure 1). By focusing on optimizing conditional probabilities through the analysis of mistakes, we enhance the model's ability to generate safe, coherent, and contextually appropriate responses. This is why mistake analysis benefits model alignment. See Section 4.2 and Appendix D for more details.
>
>
> **Q1: Resilience against adversarial instruction attacks.**
>
> Recent studies [3, 4] demonstrate that even well-aligned models, such as OpenAI's GPT-4 and Anthropic's Claude2, are vulnerable to adversarial instruction attacks, presenting a challenge distinct from aligning with human values. Our research primarily focuses on aligning LLM with human values. We have observed the effectiveness of our method in defending against delicately designed instruction attacks, such as "Goal Hijacking" with superior generalization ability (see Section 5.2). This suggests the potential effectiveness of our method in enhancing resilience against adversarial attacks. Further investigation will be considered in our future work.
>
> **Q2-3: The quality or diversity of mistake analysis for alignment improvements and more ablation study.**
>
> **(1) Quality of mistake analysis.** Yes, the quality of mistake analysis significantly impacts alignment performance. As Table 1 shows, using GPT-3.5-turbo for mistake analysis results in better alignment than using Alpaca-7B itself (see rows 7 and 8). Additionally, guided analysis outperforms unguided analysis, as shown in Table 3 (rows 3 and 4).
>
> **(2) More ablation study on mistake analysis.** In Section 5.3, we present a comprehensive ablation study on the essential components of mistake analysis, including (i) quality of mistake analysis (ii) quantity of mistake analysis and (iii) the source of responses for constructing analysis data. We find that high-quality analysis, whether guided or from advanced models like GPT, leads to better alignment. On the other hand, increasing the volume of analysis data does not always improve results. We observe a decrease in performance when multiple analyses are applied to the same instructions, possibly due to conflicting interpretations in the analyses of the same instruction. Furthermore, we observe that the alignment effectiveness is maximized when the bad cases are generated by the model itself. See Section 5.3 for detailed experimental results.

---

> > ### Author Response · Authors · 2023-11-15
> > **Response to Reviewer FapY (2/2)**
> >
> > **Q5: Safety and ethics of induced mistakes during training.**
> >
> > As reported in the 6th row of Table 4, the success ratio of the guided mistake induction exceeds 72%, which has empirically been sufficient for subsequent mistake analysis. Furthermore, we conduct guided mistake analysis, requiring the model to analyze "why the answer is potentially harmful, unethical, or offensive". This guidance aids the model in better identifying potential mistakes, as shown in Figure 8 in the Appendix. While there is some inevitable noise in such a self-alignment process, our experimental results, presented in Tables 1 and 2, demonstrate a significant improvement in the overall harmlessness rate of the models. This improvement may be attributed to the intuition that discrimination (*i.e.*, recognizing harmful responses) is simpler than generation (*i.e.*, generating harmless responses). The inherent discriminatory ability of LLMs can thus be leveraged to enhance their generative capabilities.
> >
> > [1] SAFETYPROMPTS Benchmark. http://115.182.62.166:18000/.
> >
> > [2] Sun H, Zhang Z, Deng J, et al. Safety assessment of Chinese large language models. arXiv preprint arXiv:2304.10436, 2023.
> >
> > [3] Carlini N, Nasr M, Choquette-Choo C A, et al. Are aligned neural networks adversarially aligned?. arXiv preprint arXiv:2306.15447, 2023.
> >
> > [4] Zou A, Wang Z, Kolter J Z, et al. Universal and transferable adversarial attacks on aligned language models. arXiv preprint arXiv:2307.15043, 2023.

---

> ### Author Response · Authors · 2023-11-19
> **Response to Reviewer FapY**
>
> Dear Reviewer FapY,
>
> We are grateful for the time and expertise you have contributed to the review of our manuscript. Considering your valuable suggestions, we have implemented extensive revisions to our paper. These revisions primarily include:
>
> 1. A detailed justification of the generalization capabilities of LLMs aligned with mistake analysis;
>
> 2. An in-depth theoretical discussion of how mistake analysis contributes to better alignment;
>
> 3. A discussion on the quality and diversity of mistake analysis in enhancing alignment, supplemented with additional ablation studies.
>
> We trust these changes effectively address the issues you raised. As the deadline for the discussion period draws near, we eagerly anticipate your further insights and remain dedicated to an ongoing constructive dialogue.
>
> Best regards,
>
> Authors of Paper 156

---

### Official Review · Reviewer_sdVt · 2023-11-08

**Soundness:** 2 fair
**Presentation:** 3 good
**Contribution:** 3 good
**Rating:** 5
**Confidence:** 3

**Summary:**

This paper proposes a supervised fine-tuning (SFT) method for alignment, which uses the mistake generated by LLMs themselves to help self-criticism. The authors first try to empirically demonstrate the relationship between generation and discrimination. Then, they present a method consisting of three stages including guided response generation, guided analysis generation, and unguided analysis fine-tuning. Experimental results show that the proposed method can outperform conventional alignment techniques for safety instruction following.

**Strengths:**

1. The idea which uses self-generated mistake analysis to help alignment is intuitive and interesting.

2. Experimental results show the effectiveness of the proposed method for alignment in the safety scenario.

**Weaknesses:**

1. The claim about generation and discrimination in Section 3.1 is confusing. The authors use GPT-4 scores (plus human verification) on generated responses / analysis to demonstrate that discrimination is easier than generation. But in my view, discrimination requires to output a definite classification result (such as safe or unsafe), whose performance can be easily assessed via objective metrics like accuracy. The analysis may help the model to conduct discrimination. But its generation quality is not equal to discrimination results, especially the quality is measured via GPT-4 in a “continuous” scale of 1-10.

2. The derivation in Equation (2) of Section 4 is not rigorous. The authors get a conclusion that $p(T|Y,X)$ is proportional to $p(Y|X,T)$. This indicates that the gap between $\log p(T|Y,X)$ and $\log p(Y|X,T)$ is a scalar. Following this, if we train a safety discrimination model with the loss of $\log p(T|Y,X)$, we can directly obtain a safe response generation model. This is counter-intuitive and needs more explanations. I’m afraid that the authors miss some important assumptions in their derivation.

3. In Section 5.1, the authors only use Alpaca as their base model in the experiment. Since Alpaca is a weak base model, improving its performance may be relatively easy. I wonder whether the proposed method can still work well on stronger base models such as Vicuna.

**Questions:**

I have included my questions in the weaknesses part.

---

> ### Author Response · Authors · 2023-11-15
> **Response to Reviewer sdVt (1/2)**
>
> Thank you for your valuable and constructive reviews. Here we address your questions one by one.
>
> **W1: Justification for comparing LLM's "discrimination" ability with generation" ability.**
>
> **(1) Definitions of generalization and discrimination.** Thank you for your valuable feedback. To clarify, in our paper, "generation" is defined as the LLM's ability to produce helpful and harmless responses to user instructions, while "discrimination" refers to its capacity to identify potential errors in responses, which requires not only a binary classification between "harmful" and "harmless" but also to encompass a more nuanced analysis of the response quality, as illustrated in Figure 7 of the Appendix. Therefore, the continuous score is considered as a better metric than the simple binary classification accuracy here. The preliminary aims to show the observation that LLMs are generally more adept at identifying potential mistakes than directly generating high-quality responses. This distinction is vital for the subsequent method design, where we use self-critique data to enhance the model's generation capabilities.
>
> **(2) Experiments of binary harmful/harmless classification.** However, we acknowledge your concern about regarding the potential oversight of the binary classification accuracy. Following your suggestions, we conduct an additional experiment evaluating the LLM's ability to discern the harmfulness of responses. Specifically, we utilize the test set of PKU-SafeRLHF dataset, each instruction paired with a response and the corresponding binary harmful or harmless tag, and then instruct LLMs to distinguish whether the provided response is harmful or not. The accuracy is reported in Table R1. The LLM's accuracy in identifying the harmfulness of responses improves from 54.5% for vanilla Alpaca-7B to 72.6% after applying our method, revealing the significant improvement of discrimination ability.
>
> ▾ Table R1. **Comparison between discrimination ability with respect to the binary classification accuracy on the test set of the PKU-SafeRLHF dataset.**
>
> |              | Vanilla Alpaca |           SFT | Ours |
> | :--------: | :-----------------: | :-------------: | :-----: |
> | Accuracy (%) | 54.5           | 54.9 (+0.4) | **72.6 (+18.1)** |
>
> **W2: More explanation about Equation (2) which implies a direct link between training a safety discrimination model and obtaining a safe response generation model.**
>
> Thanks for your valuable comment. We have revised the deviation of Equation (2) in Section 4 and Appendix D. In summary, your observation about the counter-intuitive nature of directly obtaining a safe response generation model from a safety discrimination model is astute when only mistake analysis data is utilized in fine-tuning. However, this scenario is different from the discussion of Equation (2), where the mistake analysis data is incorporated together with the normal SFT data, as discussed in the **"Step 2: Guided analysis generation"** paragraph of Section 4.1.
>
> Let us start with the Bayes' Theorem $p(T|Y, X) = [p(Y|X, T)p(X|T)p(T)]/[p(Y|X)p(X)]$, which simplifies to $p(T|Y, X) \propto p(Y|X, T)$ under the assumption that $p(Y|X)$ remains relatively stable during the fine-tuning process. To maintain the model's capability to produce a response given an instruction (*i.e.*, preserving $p(Y|X)$）, we combine original SFT data, both helpful ($D_{\text{helpful}}$) and harmless ($D_{\text{harmless}}$), with mistake analysis data during fine-tuning, rather than solely relying on the mistake analysis data. Therefore, Equation (2) actually aims to explore how the additional mistake analysis data, along with the original SFT instruction-response pairs, can enhance alignment performance.
>
> Moreover, our ablation study in Section 5 demonstrates that an excess of mistake analysis data, particularly when it doubles the SFT data, actually leads to poorer performance (3rd vs. 5th rows), which indicates a nuanced relationship between different types of data in fine-tuning and the model's ability to balance between generation and discrimination tasks effectively. We hope this can clarify the assumption and methodology underlying our approach and address your concerns.

---

> ### Author Response · Authors · 2023-11-15
> **Response to Reviewer sdVt (2/2)**
>
> **W3: Justification about whether the method effective on the weak Alpaca model in Section 5.1 is also applicable to stronger models like Vicuna.**
>
> **(1) Reason for choosing Alpaca.** We select Alpaca-7B as the baseline in Section 5.1 due to its absence of prior safety alignment. This characteristic renders it an ideal candidate for evaluating the effectiveness of alignment methods for an unaligned LLM from scratch. Our results, as elaborated in Table 1 and Figures 12-13, demonstrate that even unaligned LLMs like Alpaca-7B can effectively leverage  mistake analysis to enhance their safety instruction-following capabilities.
>
> **(2) Effectiveness on SOTA models.** Moreover, to address concerns about the method's effectiveness on stronger, safety-aligned models, we have expanded our experiments to include ChatGLM-6B, the highest-ranked open-source model on the C-Eval leaderboard [1]. As shown in Table 2 and Section 5.2, after incorporating the mistake analysis to defend against a novel type of instruction attack known as "Goal Hijacking", ChatGLM ranks the second on the SAFETYPROMPTS benchmark [2, 3]. Notably, it is only surpassed by OpenAI's GPT-3.5 in defending against the "Goal Hijacking" attack, while concurrently preserving its helpfulness performance.
>
> **(3) Additional experiments on Vicuna.** In line with your suggestion, we further conduct experiments on Vicuna-7B (v1.1) following the exact setting as in Section 5.1. As reported in Table R2, our mistake analysis achieves consistent improvement regradless of the baseline models.
>
> ▾ Table R2. **Comparative results of LLM alignment across various methods on Vicuna-7B.**
>
> | Method | Helpful Score | Harmless Score | Harmless Rate (%) | Harmless Helpful |
> | :------: | :-------: | :----: | :--------: | :----: |
> | Vicuna (vanilla) | 7.17          | 9.62           | 95.9              | 8.85 |
> | SFT | **7.33**      | 9.63           | 96.1              | 8.84 |
> | Ours | 7.30          | **9.74**          | **97.1**              | **8.90** |
>
> [1] C-Eval: A Multi-Level Multi-Discipline Chinese Evaluation Suite for Foundation Models. https://cevalbenchmark.com/.
>
> [2] SAFETYPROMPTS Benchmark. http://115.182.62.166:18000/.
>
> [3] Sun H, Zhang Z, Deng J, et al. Safety assessment of Chinese large language models. arXiv preprint arXiv:2304.10436, 2023.

---

> ### Author Response · Authors · 2023-11-19
> **Response to Reviewer sdVt**
>
> Dear Reviewer sdVt,
>
> We sincerely appreciate the time and expertise you have invested in reviewing our manuscript. Following your insightful comments, we have undertaken substantial revisions, including:
>
> 1. Providing a thorough justification for comparing the "discrimination" and "generation" abilities of LLMs;
>
> 2. Expanding explanations regarding Equation (2) in the main text;
>
> 2. Conducting additional experiments with more baselines such as Vicuna.
>
> We believe these modifications comprehensively address your previous concerns. With the discussion period deadline nearing, we eagerly await your further feedback and remain fully engaged in ongoing dialogue.
>
> Best regards,
>
> Authors of Paper 156

---

> > ### Comment · Reviewer_sdVt · 2023-12-04
> > **Response to Authors**
> >
> > Thanks for your detailed responses. I have read the rebuttal and the revised manuscript. The confusion about discrimination and generation has been mostly clarified. The experiment on Vicuna is also appreciated.
> >
> > About the derivation in Section 4.2, I am not sure whether the assumption ($X$ is independent with $T$) is proper in this setting. In my view, the sampling space of instructions may be still restricted by the harmful tag (e.g., we cannot sample an instruction with explicit toxic words when $T=\text{Harmless}$). Nevertheless, I agree with other reviewers that the idea is simple, efficient, and interesting. I will respect the decision from the AC and I'm surely OK if this paper is accepted.

---

### Author Response · Authors · 2023-11-15
**To All Reviewers**

We thank all reviewers for their time, insightful suggestions, and valuable comments. We are glad that ALL reviewers find our work **"novel and interesting with strong experiment results"**, and will be **"a novel addition to the pantheon of LLM fine-tuning techniques"** (Reviewer 5o83) and **"hint for bigger impact and larger application"** (Reviewer xbbn).

Below, we respond to each reviewer’s comments in detail. We have also revised the main paper and the appendix according to the reviewers’ suggestions. The main changes are listed as follows:

**In the main paper:**

1. Definitions of LLM's "discrimination" and "generation" abilities adopted in this work in Section 3.1.
2. Additional experiments of baseline methods with Alpaca-induced mistakes in Table 1.
3. Revise inappropriate wording and typos for better readability.

**In the appendix:**

1. Additional experiments of baseline methods with ChatGLM-induced mistakes in Table 6.
2. Detailed explanation of Equation (2) in Appendix D.
3. Experimental verification of Equation (2) via harmless binary tag prediction in Appendix D.
4. More qualitative comparison for safety instruction-following in Figure 9-11, and mistake analysis in Figure 12-14.

Note that we mark all the textual revision in blue and Table revision in yellow. We hope that our efforts can address the reviewers’ concerns well. Thank you very much again!

Best regards,

Paper 156 Authors

---

### Author Response · Authors · 2023-11-23
**Request for Review Feedback**

Dear All Reivewers,

As the deadline for the review process draws near, we eagerly anticipate your valuable feedback. We understand the various responsibilities you handle and stand ready to address any outstanding queries regarding our draft.

Please do not hesitate to reach out if you require further information or clarification about our work.

Thank you in advance for your time and dedication.

Best Regards,

Paper 156 Authors

---

### Meta-Review · Area_Chair_zT4h · 2023-12-06

**Metareview:**

This paper presents a supervised fine-tuning (SFT) method for aligning language models with human values. The proposed method utilizes mistakes generated by the language models themselves for self-criticism. By generating harmful responses and analyzing their own mistakes, the models learn to improve their alignment with human values. Experiments demonstrate the effectiveness of the method in improving safety instruction following and outperforming conventional alignment techniques.

Strengths

- Intuitive and interesting approach of leveraging self-generated mistakes for alignment.
- Strong experimental results showing improved harmlessness and helpfulness.
- Efficient defense against new attacks by learning from representative mistakes.

Weaknesses

- Confusing claim and derivation in Equations (2), requiring further explanation.
- Limited experimental evaluation, analysis
- Potential for generalization issues beyond training distributions.
- Difficulty in controlling mistake induction to cover all failure modes.
- Models may still make mistakes during inference.

**Justification For Why Not Higher Score:**

There are few issues in confusing claim and derivation, limited experimental analysis, and having no guarantee on the failure mode coverage. I would encourage the authors to improve the presentation further.

**Justification For Why Not Lower Score:**

The approach is intuitive and effective in leveraging self-generated mistakes for alignment. Emiprical results are strong.

---

### Decision · Program_Chairs · 2024-01-16

Accept (poster)